# Stochastic Model for a Piezoelectric Energy Harvester Driven by Broadband Vibrations

**DOI:** 10.3390/e26121097

**Published:** 2024-12-14

**Authors:** Angelo Sanfelice, Luigi Costanzo, Alessandro Lo Schiavo, Alessandro Sarracino, Massimo Vitelli

**Affiliations:** 1Department of Mathematics and Physics, University of Campania “Luigi Vanvitelli”, Viale Lincoln 5, 81100 Caserta, Italy; angelo.sanfelice@unicampania.it; 2Department of Engineering, University of Campania “Luigi Vanvitelli”, Via Roma 29, 81031 Aversa, Italy; luigi.costanzo@unicampania.it (L.C.); alessandro.loschiavo@unicampania.it (A.L.S.); massimo.vitelli@unicampania.it (M.V.)

**Keywords:** piezoelectric energy harvester, stochastic models, non-equilibrium dynamics

## Abstract

We present an experimental and numerical study of a piezoelectric energy harvester driven by broadband vibrations. This device can extract power from random fluctuations and can be described by a stochastic model, based on an underdamped Langevin equation with white noise, which mimics the dynamics of the piezoelectric material. A crucial point in the modelisation is represented by the appropriate description of the coupled load circuit that is necessary to harvest electrical energy. We consider a linear load (resistance) and a nonlinear load (diode bridge rectifier connected to the parallel of a capacitance and a load resistance), and focus on the characteristic curve of the extracted power as a function of the load resistance, in order to estimate the optimal values of the parameters that maximise the collected energy. In both cases, we find good agreement between the numerical simulations of the theoretical model and the results obtained in experiments. In particular, we observe a non-monotonic behaviour of the characteristic curve which signals the presence of an optimal value for the load resistance at which the extracted power is maximised. We also address a more theoretical issue, related to the inference of the non-equilibrium features of the system from data: we show that the analysis of high-order correlation functions of the relevant variables, when in the presence of nonlinearities, can represent a simple and effective tool to check the irreversible dynamics.

## 1. Introduction

Energy harvesting is the process of capturing and storing energy from various environmental sources and represents an important topic for both experimental and theoretical studies [1]. From the technological perspective, this field has seen significant advancements in recent years, leading to the realisation of several devices, such as photovoltaic cells, wind turbines, piezoelectric devices, thermoelectric generators, and electromagnetic energy harvesters, designed to collect energy from the environment [2,3,4].

A central issue in this framework is certainly represented by the maximisation of the extracted power. Indeed, depending on the specific design and on the parameters of the considered device, energy sources can be exploited in more efficient ways. Such a tuning of the relevant parameters can be worked out more easily if a theoretical model can be developed that can describe the real system. Since in many cases energy harvesters work under random and uncontrolled conditions, due to the unpredictable nature of the environmental energy sources, stochastic processes can play a significant role in the modelling of these systems. In particular, analytical and numerical studies of simplified models can be useful to design more robust energy harvesters that can adapt to the inherent variability of their energy sources, maximising energy capture. Let us also mention that, from a more general and theoretical perspective, the problem of rectifying random fluctuations is studied within the theory of Brownian (or molecular) motors [5], also known as ratchet models, where the presence of a spatial asymmetry coupled with non-equilibrium conditions allows one to extract directed motion from unbiased fluctuations [6,7].

Among the several mentioned physical mechanisms exploited to harvest energy from the environment, we focus here on piezoelectric materials [8]. These have the properties to convert mechanical stress into electrical energy, making them good candidates for vibration energy harvesting applications. Indeed, the electrical currents generated when the material is deformed can be collected by suitable load circuits and electrical power can be obtained. Devices based on this mechanism are usually used to feed small sensors, for instance in wireless sensor networks. These kinds of energy harvesters are generally constituted by a piece of piezoelectric material whose tip mass is subjected to some vibration and is also electromechanically coupled with the current flowing in the electric circuit. Although piezoelectric vibration harvesters are typically studied in sinusoidal conditions, at frequencies within their resonance band [9,10], as mentioned above it is also important to consider broadband vibrations, which can be modelled as white noise [11,12,13,14,15,16,17]. From a more theoretical perspective, stochastic driving makes this kind of system an interesting instance where results from the general theory of stochastic thermodynamics [18] can be applied, as, for instance, discussed in [15]. This theory represents an attempt to generalize the concepts of standard thermodynamics, such energy, heat, and entropy, to systems where fluctuations play a central role and cannot be neglected. For instance, in [15], it is shown how a fluctuating power can be defined according to the prescriptions of stochastic thermodynamics for this kind of system. Moreover, as detailed below in the model equations, the coupling between tip mass velocity and electrical current introduces a feedback mechanism in the system that can be represented as memory effects [19].

We will present results from experiments performed on the piezoelectric harvesters with two different setups for the load electrical circuit: (I) a linear configuration, which allows for analytical computations, as previously studied in [15], in order to fix the main relevant parameters of the model; (II) a nonlinear configuration, featuring a diode bridge as the output circuit. We present stochastic models that can capture the essential physical processes and mechanisms underlying the energy conversion in these configurations, taking into account various parameters, such as the mechanical properties of the piezoelectric harvester and the electrical characteristics of the harvesting circuit. These models are mainly based on an underdamped Langevin equation, which describes the dynamics of the tip mass of the piezoelectric material, subjected to the viscous drag of the air and confined by an elastic potential. The tip mass velocity is also electromechanically coupled with the current flowing in the electric circuit, as described in the following in detail. Moreover, the piezoelectric material is connected with a shaker, which represents the source of vibrations which are converted to electrical power. In particular, regarding the nonlinear configuration, we will show that a simplified effective model for the diode bridge rectifier is sufficient to reproduce the average extracted power.

Moreover, the system under study allows us to address an interesting issue related to the inference of the non-equilibrium properties of the system from a temporal series of data [20]. We will show that, at variance with the linear case discussed in [17], in the nonlinear configuration the time asymmetry can be revealed by the analysis of high-order correlation functions of a single variable.

This paper is organised as follows. In Section 2, we describe the experimental setting, and provide details on the two configurations considered. In Section 3, we present the stochastic theoretical models, based on the Langevin equation. In particular, we propose an effective equation for the modelling of the diode bridge rectifier. We compare analytical and numerical results with the experimental data, finding very good agreement in a wide range of parameters. In Section 3.3, we discuss the problem of inferring the time asymmetry of the system from data analysis. Finally, in Section 4 we present some conclusions and perspectives for future works.

## 2. Experimental Setup

The employed experimental setup is shown in the photo reported in Figure 1. In particular, the considered harvester is the commercial piezoelectric device MIDE PPA-4011 (by MIDE Technology, Woburn, MA, USA). The harvester was mounted in a cantilever resonant structure and placed on a shaker, the VT-500 by Sentek (by Sentek Dynamics, Santa Clara, CA, USA), which was used as the source of the desired vibrations. The shaker driving current was provided by a Power Amplifier LA-800 (by Sentek Dynamics, Santa Clara, CA, USA) whose control signal was generated by closed-loop vibration control implemented by a Crystal Instruments Spider-81 (by Crystal Instruments, Santa Clara, CA, USA) measuring the shaker acceleration by means of an accelerometer Dytran 3055D2 (by Dytran (HBK), Chatsworth, CA, USA).

Schematic and circuital representations of the experimental setup showing the different considered harvester loads are reported in Figure 2 and Figure 3. In particular, the piezoelectric harvester was forced by broadband vibrations of Gaussian type, with sampling rate $f_{s}=5$ kHz and different standard deviations (0.8 g and 1 g, where g is gravity acceleration). As shown in Figure 2, firstly a linear resistive load was considered and the voltage $v_{p}$ across the load resistance *R* was measured and recorded for different values of *R*. The second considered harvester load was a nonlinear load consisting of a diode bridge rectifier. Such a kind of circuit is typically employed in piezoelectric vibration energy harvesting applications, in cases with both laboratory prototypes and commercial devices [21,22,23,24,25], with the aim of carrying out the AC/DC conversion, which is necessary for supplying electronic DC loads (like sensors of a wireless sensor network). As shown in Figure 3, the diode bridge rectifier (made of four 1N4148 diodes) is connected to the parallel between a capacitor ($C_{DC}=100\;\mu$F) and a resistor with resistance *R*. In this case, the voltage $v_{p}$ at the input of the diode bridge rectifier and the voltage $v_{DC}$ across the load resistance *R* were measured and recorded for different values of *R*.

## 3. Theoretical Model

To describe the experimental setup, we consider the following stochastic model: (1)$$\begin{array}{ccc} \dot{x} & = & v, \end{array}$$(2)$$\begin{array}{ccc} M\dot{v} & = & -K_{s}x-\gamma v-\theta v_{p}+M\xi , \end{array}$$(3)$$\begin{array}{ccc} C_{p}{\dot{v}}_{p} & = & \theta v-i_{p}, \end{array}$$(4)$$\begin{array}{ccc} i_{p} & = & f(v_{p}), \end{array}$$
where ξ is white noise with zero mean and correlation $\langle \xi \left(t\right)\xi \left(t^{\prime }\right)\rangle =2D_{0}\delta \left(t-t^{\prime }\right)$. In the above equations, *x* represents the displacement of the tip mass *M*, *v* represents its velocity, γ represents the viscous friction due to air, $K_{s}$ represents the stiffness of the cantilever in the elastic approximation, $v_{p}$ represents the voltage across the load resistance *R*, $C_{p}$ represents the effective capacitance in the circuit, θ represents the electromechanical coupling factor of the transducer, $i_{p}$ represents the current flowing in the electrical circuit, and $f(v_{p})$ represents the characteristic current-voltage of the electrical load connected with the piezoelectric harvester. Thermal fluctuations on the tip mass are too small to affect its motion and are neglected.

The explicit forms of the function $f(v_{p})$ corresponding to the two considered experimental configurations are
(5)$$f\left(v_{p}\right)=\frac{v_{p}}{R},$$
for the linear case, which we denote by Configuration (I), and
$$\begin{array}{ccc} f(v_{p}) & = & \left(I_{k0}+Gv_{DC}\right)\left(e^{\frac{v_{p}-v_{DC}}{2\eta V_{T}}}-e^{\frac{-v_{p}-v_{DC}}{2\eta V_{T}}}\right), \end{array}$$
(6)$$\begin{array}{ccc} C_{DC}\frac{dv_{DC}}{dt} & = & -\frac{v_{DC}}{R}+\left(I_{k0}+Gv_{DC}\right)\left(e^{\frac{v_{p}-v_{DC}}{2\eta V_{T}}}+e^{\frac{-v_{p}-v_{DC}}{2\eta V_{T}}}-2\right), \end{array}$$
for the nonlinear (diode bridge rectifier) case, denoted as Configuration (II). Here, η=1.94 for the considered diodes and $V_{T}=25$ mV is the thermal voltage.

### 3.1. Linear Load

Case (I) has been carefully studied in previous works [15,16,17] and, due to the linear nature of the model, allows for analytical treatment. It is considered here in order to fit some of the model parameters, which are then kept fixed in setup (II). We report the explicit expression for the average output power extracted by the harvester, which corresponds to the heat dissipated into the load resistance *R* per unit time
(7)$$P_{harv}^{\left(I\right)}=\frac{1}{R}\langle v_{p}^{2}\rangle =\frac{D_{0}M^{2}R\theta^{2}}{M\left(\gamma +R\theta^{2}\right)+C_{p}R\gamma \left(C_{p}K_{s}R+\gamma +R\theta^{2}\right)},$$
where the symbol 〈⋯〉 denotes an average over noise in the stationary state. Details on the analytical solution of model (I) can be found in [15].

The noise amplitude $D_{0}$ is related to the shaker acceleration *a* and to the sampling rate 1/Δt of the input signal, $D_{0}=a^{2}\Delta t/2$, where $\Delta t=1/f_{s}=0.0002$ s. The ratio of parameters $K_{s}/M$ is fixed by the characteristic frequency of the device, with is $\sqrt{K_{s}/M}=2\pi \times 100$ Hz. The capacitor $C_{p}∼410$ nF is measured by using an LCR meter U1733C by Keysight Technologies, Colorado Springs, CO, USA. The other parameters are fitted to the experimental data exploiting the analytical expression (Equation 7) as a function of the load resistance *R*. For the case where $a=0.8\times 9.81\;{m/s}^{2}$, we obtain the following values: M=0.0112±0.0005 Kg, θ=0.0172±0.0005 N/V, and γ=0.660±0.005 Kg/s. This set of parameters is used also for other values of the shaker accelerations used in the experiments, $a=1.0\times 9.81\;{m/s}^{2}$. In Figure 4, we report the experimental data and the analytical curve for case (I). We observe a non-monotonic behaviour of the extracted power as a function of the load resistance, with a peak corresponding to the optimal load $R^{*}∼3000–4000\;\Omega$. As predicted by analytical Formula (Equation 7), this value is independent of the forcing acceleration, as shown in the right panel of Figure 4. In this figure, we also observe that the parameter values reported above (fitted to the case where a=0.8 g) describe quite well the behaviour for a different value of the acceleration (a=1.0 g).

As already shown in previous works [15], the efficacy of the linear model in describing the real system is not limited to mean values of the relevant quantities, but also extends to the fluctuations in the voltage $v_{p}$, as reported in Figure 5. Here, we show the histograms of the measured values of $v_{p}$ in experiments and in numerical simulations for a fixed value of the load resistance R=1500Ω, for the two values of the considered acceleration, which are in very good agreement. As expected from the linearity of the model, these distributions show a Gaussian shape. The analytical expression for the variance of the Gaussian as a function of the model parameters is reported in [15]. For other values of the load resistance, we find similar behaviours.

### 3.2. Nonlinear Load

An analytical treatment is not possible for the model of Configuration (II), due to the strong nonlinearities appearing in the equations. Therefore, here we focus on numerical simulations in order to investigate the accuracy of the proposed modelling, in particular for what concerns the description of the diode bridge. Here, we simplify the treatment, introducing two effective parameters, $I_{k0}$ and *G*, that appear in Equation (Equation 6). The effective values of these parameters used in numerical simulations are $I_{k0}=9\times 10^{-9}$ A and $G=-133\times 10^{-11}\;\Omega^{-1}$.

As first, we consider the voltage $v_{p}$ and compare the probability distributions measured in experiments with those obtained from numerical simulations. The good agreement between experiments and simulations for different values of the load resistance *R* observed in Figure 6, Figure 7 and Figure 8 shows that fluctuations are well described by the nonlinear model. In particular, we note that these distributions have a shape very different from a Gaussian, due to the nonlinearity of the system. These differences appear more pronounced at small values of *R* and are characterised by large tails. Similar behaviours are observed for both the values of the considered acceleration and for other values of the resistance *R*.

We then consider the quantity $v_{DC}$, which is relevant for the harvested power through the expression
(8)$$P_{harv}^{\left(II\right)}=\frac{\langle v_{DC}^{2}\rangle }{R}.$$ In order to test the accuracy of the model, we first consider the average value $\langle v_{DC}\rangle$, which is reported in Figure 9 for different values of the load resistance. Good agreement is found between experiments and numerical simulations. We observe an increasing behaviour with *R*. This is expected since, the more *R* increases, the closer the system approaches open-circuit operating conditions.

We then consider the harvested power, which is reported in Figure 10, and which involves the fluctuations $\langle v_{DC}^{2}\rangle$. This is the most important quantity, because it represents the energy that can be practically used to power other devices. Again, with the considered parameters, good agreement is found, in the whole range of values of *R*, between the numerical simulations and experimental results. We observe a non-monotonic behaviour characterised by a maximum for the optimal load resistance $R^{*}∼30,000\;\Omega$. We note that the maximum extracted power is smaller with respect to the linear case. However, as mentioned above, this kind of load circuit is necessary in practical applications to carry out the AC/DC conversion for supplying electronic loads.

Finally, in Figure 11 we compare the probability distributions for the quantity $v_{DC}$ measured in experiments and in simulations for R=3300Ω. We observe that, even if average and variance are well described by the model, the whole shape of the distributions is not accurately reproduced. Other values of *R* show similar behaviours. This is probably due to the simplifications introduced in the model to represent the diode bridge. Our analysis, therefore, shows that the proposed model is a good compromise between accuracy and simplicity, allowing us to reproduce the behaviour of the main quantities, without the introduction of a large number of effective parameters.

### 3.3. Time-Correlation Functions and Temporal Asymmetry

The system considered in our study allows us to also address a theoretically interesting problem, consisting of the assessment of the non-equilibrium nature of the system from partial information, namely from the measurement of only some degrees of freedom. This is a non-trivial issue, because in experiments one usually cannot directly access the dynamics of all the relevant quantities and the evaluation of the temporal (a-)symmetry of the system behaviour can be difficult. The question plays a central role in the general problem of finding a good model from data, where knowledge of the equilibrium/non-equilibrium properties of the system can provide a useful base for modelisation. Several theoretical tools can be used to assess such features, such as entropy production measurements [26], violations of the fluctuation–dissipation relations [27], and high-order correlation functions [28]. In particular, as recently discussed in [20], the Gaussian nature of the model can hide the non-equilibrium features of the dynamics when only one degree of freedom is considered, leading to the necessity of the analysis of cross-correlation between two variables to assess the non-equilibrium behaviours. This point was considered in detail in a previous paper of some of the present authors [17], where the linear model of Configuration (I) was considered, and indeed it was shown that the single measurement of the voltage $v_{p}$ was not enough to unveil the non-equilibrium system dynamics. In that case, the cross-correlation of $v_{p}$ with another variable related to the displacement of the mass tip of the piezoelectric was considered.

Here, exploiting the nonlinear components of the system in Configuration (II), we show that from the analysis of the time series of a single variable it is possible to reveal the time asymmetry of the dynamics. In particular, we define the connected three- and four-point correlation functions of the voltage $v_{DC}$ as follows:(9)$$C_{v_{DC}}^{\left(3\right)}\left(t\right)=\frac{\langle v_{DC}\left(t\right)v_{DC}{\left(0\right)}^{2}\rangle -\langle v_{DC}\rangle \langle v_{DC}^{2}\rangle }{\langle v_{DC}^{3}\rangle -\langle v_{DC}\rangle \langle v_{DC}^{2}\rangle },$$
(10)$$C_{v_{DC}}^{\left(4\right)}\left(t\right)=\frac{\langle v_{DC}\left(t\right)v_{DC}{\left(0\right)}^{3}\rangle -\langle v_{DC}\rangle \langle v_{DC}^{3}\rangle }{\langle v_{DC}^{4}\rangle -\langle v_{DC}\rangle \langle v_{DC}^{3}\rangle }.$$ The choice of these kinds of correlation functions is dictated by the observation that simple two-point autocorrelation is always time-symmetric by definition and, therefore, higher-order functions have to be taken into account. In Figure 12, we report $C_{v_{DC}}^{\left(3\right)}\left(t\right)$ (left panel) and $C_{v_{DC}}^{\left(4\right)}\left(t\right)$ (right panel) measured in experiments with R=3300Ω for a=0.8 g. Other values of parameters show similar behaviours. We clearly see the time asymmetry of the dynamics from the difference between these functions and those obtained inverting the time argument.

In order to verify in more detail how much the model can capture the non-equilibrium properties of the real system, we also compute the same high-order correlation functions from numerical simulations, for the same parameters used before. In Figure 13, we report $C_{v_{DC}}^{\left(3\right)}\left(t\right)$ (left panel) and $C_{v_{DC}}^{\left(4\right)}\left(t\right)$ (right panel), which show a qualitative behaviour similar to what we observed in the case of experimental data: the presence of a peak at small times, in particular, even if less evident with respect to the experimental data, signals the temporal asymmetry of the dynamics, and, therefore, confirms the non-equilibrium nature of the theoretical model. The quantitative disagreement is probably due to the simplifications introduced in the modelling of the diode bridge, as already discussed before.

## 4. Conclusions

This work focuses on the modelisation of a piezoelectric energy harvester driven by random broadband vibrations. We analyse data obtained in two different experimental setups: a linear configuration, characterised by a single load resistance in the output circuit, and a nonlinear one, where a diode bridge is considered. The theoretical model proposed to describe the experimental results relies on a stochastic equation, based on an underdamped Langevin equation for the tip mass dynamics of the piezoelectric material, electromechanically coupled with the output electrical circuit. As also shown in previous studies [15,16,17], the system in the linear setup is very well fitted by the theoretical model. Here, we consider this configuration to fix some of the model parameters. We then analyse the accuracy of the model when the diode bridge is studied. This nonlinear configuration is more realistic for practical implementations but cannot be treated analytically. Therefore, we perform extensive numerical simulations of the model to investigate the role of the parameters, in particular on the dependence of the extracted power on the load resistance, finding a good agreement between experiments and numerical simulations. In particular, our approach proposes a simplified model for the description of the diode circuit, which allows us to recover the experimental behaviour with a small number of parameters. The main result of our analysis consists of the observation of a non-monotonic behaviour of the extracted power as a function of the load resistance, identifying the optimal value at which the power is maximised.

We also address the more theoretical issue related to the characterisation of non-equilibrium features in the system from the analysis of a time series of a single observable variable. At variance with previous studies performed in the linear setup, we show here that, as expected from general arguments, in the case of nonlinear systems, to demonstrate a temporal asymmetry of the dynamics, it is not necessary to measure cross-correlations between two different variables, but the computation of high-order correlation functions of a single variable is sufficient.

Our study extends the validity of the proposed stochastic model to treat piezoelectric energy harvesters driven by broadband vibrations to nonlinear setups and represents a physical example where the properties of non-equilibrium fluctuations in systems with feedback dynamics can be studied.

## Figures and Tables

**Figure 1 entropy-26-01097-f001:**
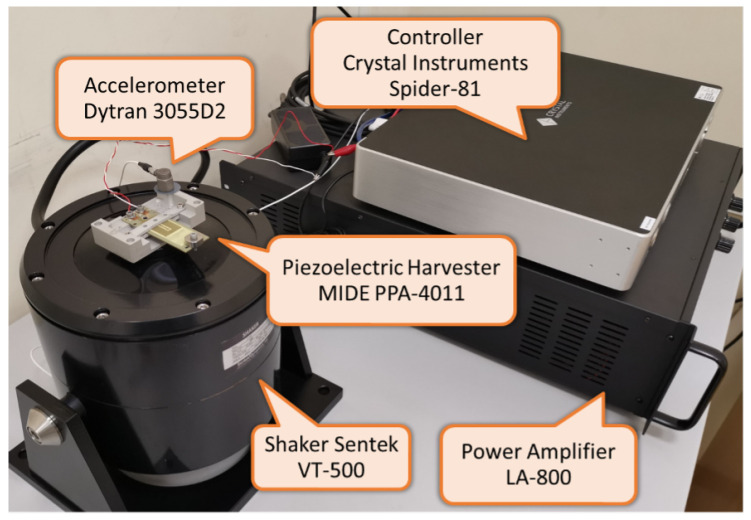
Picture of the experimental setup.

**Figure 2 entropy-26-01097-f002:**
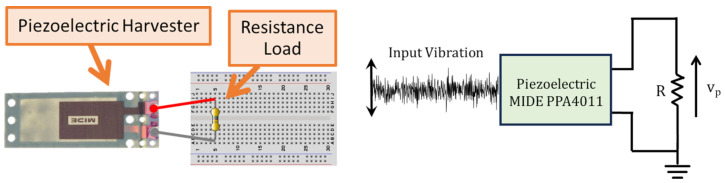
Schematic representation (**left**) and circuital representation (**right**) of the piezoelectric harvester loaded by the resistive load.

**Figure 3 entropy-26-01097-f003:**
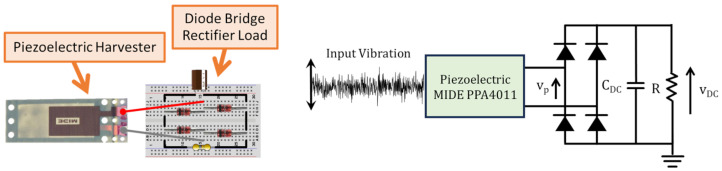
Schematic representation (**left**) and circuital representation (**right**) of the piezoelectric harvester loaded by the diode bridge rectifier load.

**Figure 4 entropy-26-01097-f004:**
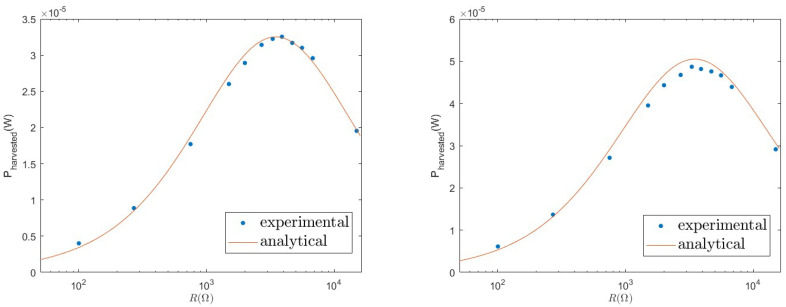
Comparison between the extracted power $P_{harv}^{\left(I\right)}$ in the linear setup (I) measured in experiments (dots) and the analytical prediction of Equation (Equation 7) (line), with the values of the parameters reported in the text. (**Left**): acceleration a=0.8 g. (**Right**): acceleration a=1.0 g. The parameters are fitted to the case where a=0.8 g.

**Figure 5 entropy-26-01097-f005:**
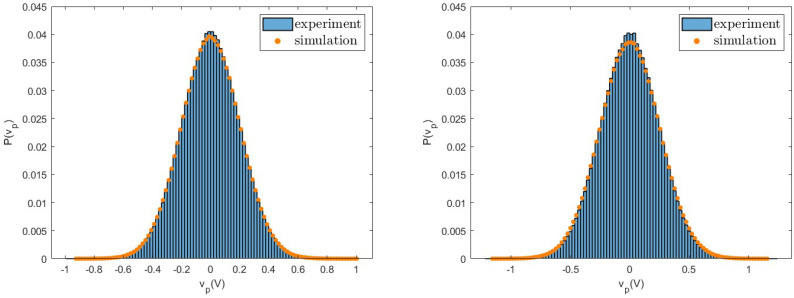
Probability distributions of the voltage $v_{p}$ measured in experiments (histogram) and in numerical simulations (dots) in the case of load resistance R=1500Ω, for Configuration (I). (**Left**): acceleration a=0.8 g. (**Right**): acceleration a=1.0 g. Other values of the resistance show similar behaviours.

**Figure 6 entropy-26-01097-f006:**
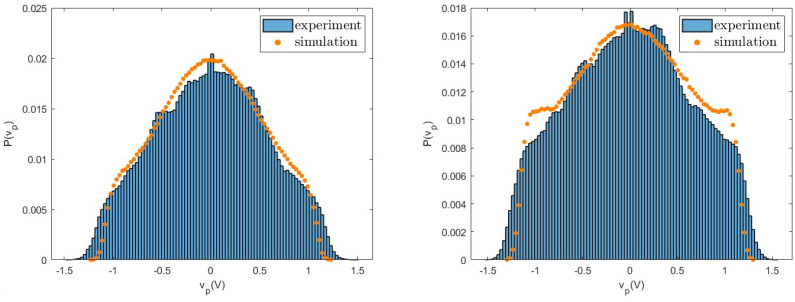
Probability distribution of the voltage $v_{p}$ measured in experiments and numerical simulations for R=3300Ω for a=0.8 g (**left**) and a=1.0 g (**right**).

**Figure 7 entropy-26-01097-f007:**
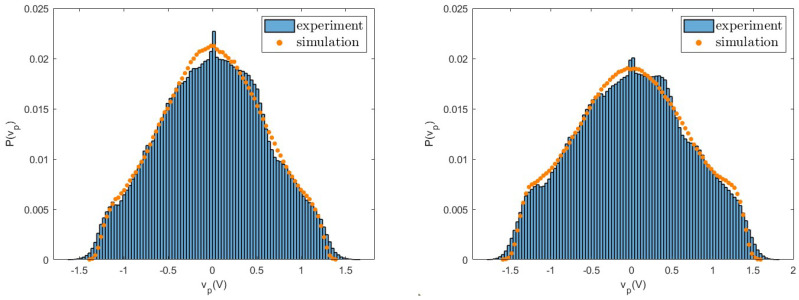
Probability distribution of the voltage $v_{p}$ measured in experiments and numerical simulations for R=20,000Ω for a=0.8 g (**left**) and a=1.0 g (**right**).

**Figure 8 entropy-26-01097-f008:**
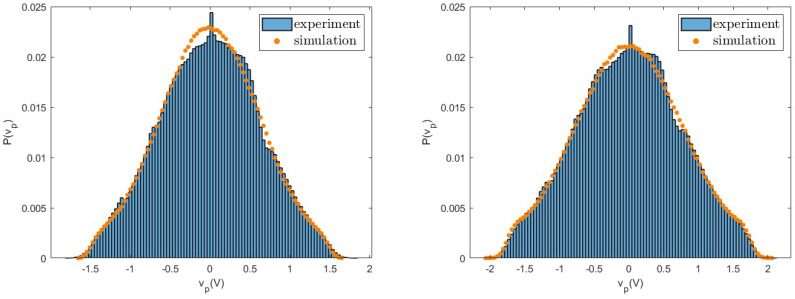
Probability distribution of the voltage $v_{p}$ measured in experiments and numerical simulations for R=100,000Ω for a=0.8 g (**left**) and a=1.0 g (**right**).

**Figure 9 entropy-26-01097-f009:**
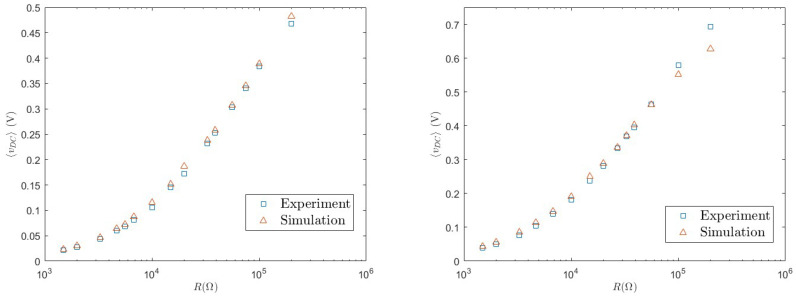
Average $\langle v_{DC}\rangle$ as a function of *R* for a=0.8 g (**left**) and a=1.0 g (**right**).

**Figure 10 entropy-26-01097-f010:**
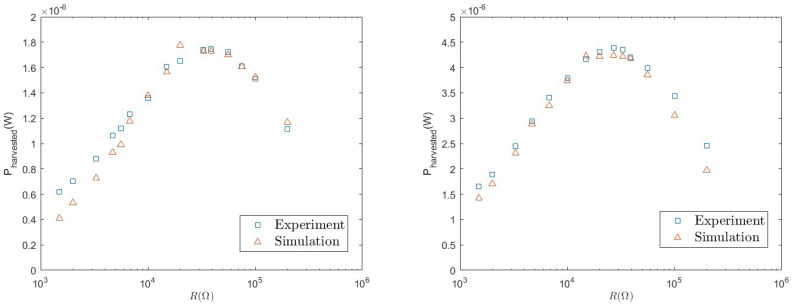
Extracted power $P_{harv}^{\left(II\right)}=\langle v_{DC}^{2}\rangle /R$ as a function of *R* for a=0.8 g (**left**) and a=1.0 g (**right**).

**Figure 11 entropy-26-01097-f011:**
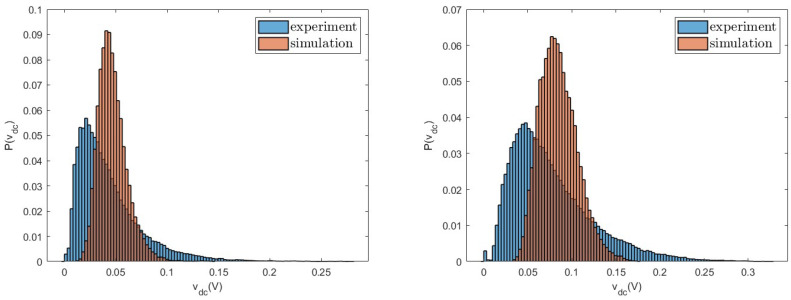
Probability distributions of the voltage $v_{DC}$ measured in experiments and numerical simulations for R=3300Ω for a=0.8 g (**left**) and a=1.0 g (**right**).

**Figure 12 entropy-26-01097-f012:**
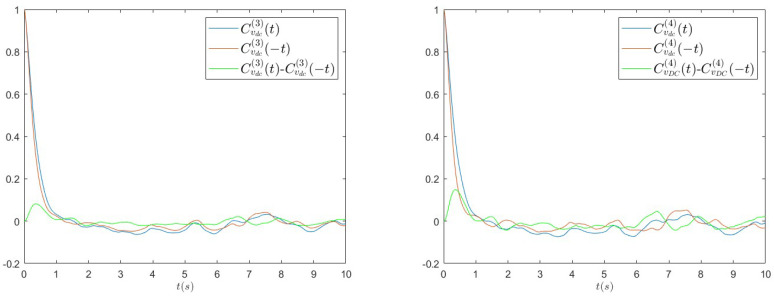
Multi-point correlation functions of the voltage $v_{DC}$, $C_{v_{DC}}^{\left(3\right)}\left(t\right)$ (left panel) and $C_{v_{DC}}^{\left(4\right)}\left(t\right)$ measured in experiments with R=3300Ω for a=0.8 g. The time asymmetry is revealed by the difference with respect to the same functions computed by inverting the time argument.

**Figure 13 entropy-26-01097-f013:**
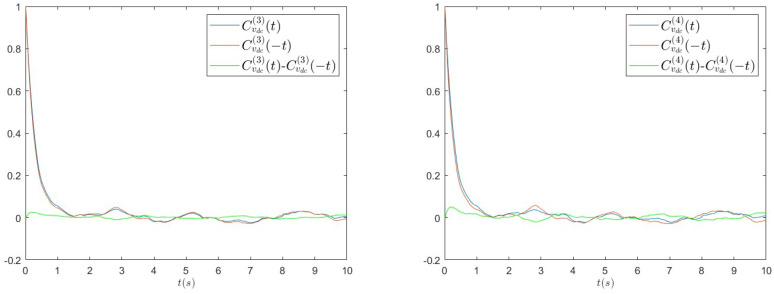
Multi-point correlation functions of the voltage $v_{DC}$, $C_{v_{DC}}^{\left(3\right)}\left(t\right)$ (left panel) and $C_{v_{DC}}^{\left(4\right)}\left(t\right)$ measured in numerical simulations with R=3300Ω for a=0.8 g. The time asymmetry is revealed by the difference with respect to the same functions computed by inverting the time argument.

## Data Availability

The original contributions presented in this study are included in the article. Further inquiries can be directed to the corresponding author.

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
