# Peer review of "Stochastic Model for a Piezoelectric Energy Harvester Driven by Broadband Vibrations"

_entropy, 2024, doi:10.3390/e26121097_

Round 1
Reviewer 1 Report
Comments and Suggestions for Authors
This paper is strongly based on two earlier works by the same authors, published between 2021 and 2022, which investigated an energy harvester device driven by broadband random vibrations using both empirical and theoretical approaches.
While all three works share a common focus on energy production and employ almost the same stochastic model proposed by the authors to interpret experimental data, they differ in two key aspects: the experimental setup (e.g. current generated by the oscillations of a magnet or a piezoelectric element and measured directly across a resistor or after diode bridge rectification) and the analytical focus in order to compare theory and experiment (looking at the average quantities or analyzing the distribution and time correlations of voltage fluctuations).
Despite these similarities, this article introduces some significant innovations compared to the previous studies.
One major novelty is the inclusion of a nonlinear term in the stochastic model, reflecting the impact of the diode bridge. This is a substantial change that considerably complicates the analytical study. So the authors had to address this by using numerical simulations.
Another key advancement is the analysis of the voltage fluctuation distribution measured across the resistor. By leveraging the non-linearity introduced by the diode bridge, the authors explore the system's irreversibility through connected three- and four-point time correlation functions of a single variable. This approach is typically infeasible in the linear Gaussian case.
In my opinion, the article effectively presents its assumptions and objectives, employs rigorous methodologies, and interprets results thoughtfully. I particularly appreciate the authors’ transparency in highlighting the discrepancies between model and experiment in Fig. 11.
That said, I believe the study could benefit from one additional element. To improve the comparison between theory and experiments, I suggest adding to Fig. 12 the connected three- and four-point correlation functions obtained from model simulations. Even if the simulated curves would differ significantly from experimental results (which seems likely given the distribution of v_{DC}​ in Fig. 11), such an addition would be valuable. It would quantify how much the model is able to capture the non-equilibrium properties of the real system, thereby providing deeper insights.
In any case, I feel that the paper meets the criteria of scientific quality required by Entropy and I have no doubt to recommend publication.
Author Response
Reply to Referee 1
We deeply thank the Referee for their accurate reading of our
manuscript and for their positive judgment on our work.
We also thank the Referee for their comment about the behavior of high
order correlation function in the numerical simulations. Following
their suggestion we have added a figure with two panels, showing the
correlation functions measured in the numerical simulations. The
qualitative behavior is similar to what we can observe from the
analysis of the experimental data, confirming the nonequilibrium
character of the model. The quantitative disagreement is probably due
to the simplifications introduced in the modeling of the diode bridge,
as already discussed in text. We have added a discussion of this point
in the revised version of the paper. Revisions are in blue in the text.
Reviewer 2 Report
Comments and Suggestions for Authors
In the manuscript "Stochastic model for a piezoelectric energy harvester driven by broadband vibrations", the authors study experimentally a system called a "piezoelectric energy harvester" - which to my understanding, is a noisy electric circuit with a diode. They measure the output power and the distribution of the voltage on the diode, as a function of the parameters of the model.
The authors first analyze a far simpler case in which the diode is replaced by a resistor - case (I). In this case, which has been studied in previous works, the system is Gaussian and the model is analytically solvable. They demonstrate good agreement between theory and experiment in this case (Figs. 4 and 5).
Next, they study the case (II) which is analytically more complicated due to the nonlinear behavior of the diode. Here they find fairly good agreement between experiment and simulations in most cases, with the exception of a fairly strong disagreement shown in Fig. 11 for the PDF of the direct-current voltage.
They also demonstrate time-reversal asymmetry by studying temporal correlations.
Speaking as a reader who is not a specialist in the specific system studied and the experimental techniques used in this work, and was not familiar with the previous works [16-18], I found it difficult to understand the motivation for studying this particular system and the significance and implications of the results. I believe that the paper is not written in a very self-contained form, and that the authors implicitly assume that the reader is already familiar with the specifics (e.g., that the reader has already read some of Refs. [16-18]). Moreover, in some places the text looks like it was copy-pasted from previous works by the authors with minor modifications (e.g., the structure of the paper is exactly like Ref. [16], and the very beginning of the Conclusion is identical), and/or reads like something that was generated by ChatGPT (e.g., the first paragraph of the introduction in which many statements about climate change etc are made, which appear to me not to be so relevant to the current work). These problems appear to be especially abundant in the abstract, introduction and conclusion.
I have also one specific comment: In the last sentence, the authors appear to claim that their system is non-Markovian. I don't see from Eqs. (1)-(6) why this is the case, as it seems like the only information necessary to evolve the system in time is its current state (where the state could be represented by the vector (x, v, v_p, v_{DC})).
To the best of my understanding, the science presented in the technical sections of this paper is probably sound and the results are likely to be novel, and perhaps of interest to researchers in the field. It would be helpful to add in these sections some brief discussions of the significance of the results and to explain them a little better.
However, I believe the abstract, introduction and most of the conclusion should be completely re-written if this paper is to be published in a scientific journal, and I would recommend simply deleting them and starting from scratch. If the presentation is thus significantly improved, after a second reading of the text I would be in a position to make an intelligent recommendation about whether this paper should be published in Entropy.
Author Response
Reply to Referee 2
We deeply thank the Referee for their thorough reading of our
manuscript and for their very useful comments that we have taken into
account in the revised version of the manuscript.
We thank the Referee for their comments on the readability of the
paper. Indeed, this gave us the opportunity to present in a more clear
and self-contained form our work. We also apologize if the
Introduction of the paper appeared too general and not focused on the
system under study: our aim was to provide a wider scope to our work,
but we completely understand the objections of the Referee. We have
rewritten the Introduction and modified the Conclusions, following
Referee's suggestions. The structure of the paper follows the
classical structure of a scientific work, where experimental setup,
theoretical model and data analysis are presented. Since this is the
forth of a series of works on the same subject, we retained the same
structure.
We also thank the Referee for their comment on the Markovianity of the
considered model. Indeed, the Referee is correct. What we meant is
that in the system there are feedback effects, because the variable
representing the tip mass velocity is coupled with other variables
and, at least in the linear case, one could write a single effective
Langevin equation for the mass velocity with a memory kernel taking
into account such coupling. We have corrected the text on this
important point.
We thank the Referee for their appreciation of our results. We added
some sentences to clarify their significance and impact in Section 2.2
and 2.3.
We hope that the revised version, much improved thanks to the comments
of the Referees, can be now suitable for publication. Revisions are in blue in the text.
Round 2
Reviewer 2 Report
Comments and Suggestions for Authors
The authors have made significant improvements, especially in the introduction and conclusions sections. It is now possible to understand it much better, and I recommend it for publication in Entropy.